# Marginals-to-Models Reducibility

**Tim Roughgarden**
Stanford University
tim@cs.stanford.edu

**Michael Kearns**
University of Pennsylvania
mkearns@cis.upenn.edu

## Abstract

We consider a number of classical and new computational problems regarding marginal distributions, and inference in models specifying a full joint distribution. We prove general and efficient reductions between a number of these problems, which demonstrate that algorithmic progress in inference automatically yields progress for "pure data" problems. Our main technique involves formulating the problems as linear programs, and proving that the dual separation oracle required by the ellipsoid method is provided by the target problem. This technique may be of independent interest in probabilistic inference.

## 1 Introduction

The movement between the specification of "local" marginals and models for complete joint distributions is ingrained in the language and methods of modern probabilistic inference. For instance, in Bayesian networks, we begin with a (perhaps partial) specification of local marginals or CPTs, which then allows us to construct a graphical model for the full joint distribution. In turn, this allows us to make inferences (perhaps conditioned on observed evidence) regarding marginals that were not part of the original specification.

In many applications, the specification of marginals is derived from some combination of (noisy) observed data and (imperfect) domain expertise. As such, even before the passage to models for the full joint distribution, there are a number of basic computational questions we might wish to ask of given marginals, such as whether they are consistent with *any* joint distribution, and if not, what the nearest consistent marginals are. These can be viewed as questions about the "data", as opposed to inferences made in models derived from the data.

In this paper, we prove a number of *general, polynomial time reductions* between such problems regarding data or marginals, and problems of inference in graphical models. By "general" we mean the reductions are not restricted to particular classes of graphs or algorithmic approaches, but show that any computational progress on the target problem immediately transfers to progress on the source problem. For example, one of our main results establishes that the problem of determining whether given marginals, whose induced graph (the "data graph") falls within some class $\mathcal{G}$, are consistent with any joint distribution reduces to the problem of MAP inference in Markov networks falling in the same class $\mathcal{G}$. Thus, for instance, we immediately obtain that the tractability of MAP inference in trees or tree-like graphs yields an efficient algorithm for marginal consistency in tree data graphs; and any future progress in MAP inference for other classes $\mathcal{G}$ will similarly transfer. Conversely, our reductions also can be used to establish negative results. For instance, for any class $\mathcal{G}$ for which we can prove the intractability of marginal consistency, we can immediately infer the intractability of MAP inference as well.

There are a number of reasons to be interested in such problems regarding marginals. One, as we have already suggested, is the fact that given marginals may not be consistent with any joint

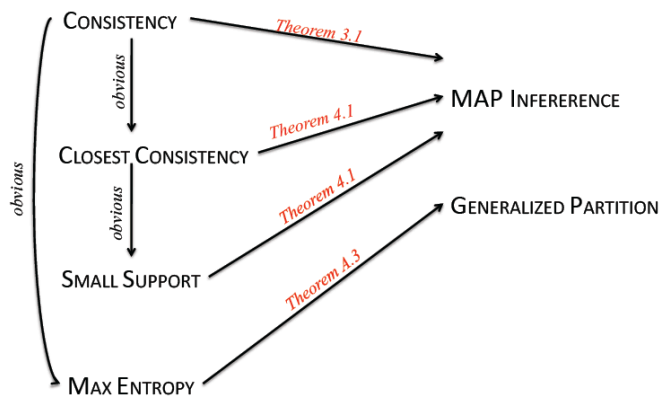

Figure 1: Summary of main results. Arrows indicate that the source problem can be reduced to the target problem for any class of graphs $\mathcal{G}$, and in polynomial time. Our main results are the left-to-right arrows from marginals-based problems to Markov net inference problems.

distribution, due to noisy observations or faulty domain intuitions,[1] and we may wish to know this before simply passing to a joint model that forces or assumes consistency. At the other extreme, given marginals may be consistent with *many* joint distributions, with potentially very different properties.[2] Rather than simply selecting one of these consistent distributions in which to perform inference (as would typically happen in the construction of a Markov or Bayes net), we may wish to reason over the entire class of consistent distributions, or optimize over it (for instance, choosing to maximize or minimize independence).

We thus consider four natural algorithmic problems involving (partially) specified marginals:

- CONSISTENCY: Is there any joint distribution consistent with given marginals?
- CLOSEST CONSISTENCY: What are the consistent marginals closest to given inconsistent marginals?
- SMALL SUPPORT: Of the consistent distributions with the closest marginals, can we compute one with support size polynomial in the data (i.e., number of given marginal values)?
- MAX ENTROPY: What is the maximum entropy distribution closest to given marginals?

The consistency problem has been studied before as the membership problem for the marginal polytope (see Related Work); in the case of inconsistency, the closest consistency problem seeks the minimal perturbation to the data necessary to recover coherence.

When there are many consistent distributions, which one should be singled out? While the maximum entropy distribution is a staple of probabilistic inference, it is not the only interesting answer. For example, consider the three features "votes Republican", "supports universal healthcare", and "supports tougher gun control", and suppose the single marginals are 0.5, 0.5, 0.5. The maximum entropy distribution is uniform over the 8 possibilities. We might expect reality to hew closer to a small support distribution, perhaps even 50/50 over the two vectors 100 and 011. The small support problem can be informally viewed as attempting to minimize independence or randomization, and thus is a natural contrast to maximum entropy. It is also worth noting that small support distributions arise naturally through the joint behavior of no-regret algorithms in game-theoretic settings [1].

We also consider two standard algorithmic inference problems on full joint distributions (models):

- MAP INFERENCE: What is the MAP joint assignment in a given Markov network?
- GENERALIZED PARTITION: What is the normalizing constant of a given Markov network, possibly after conditioning on the value of one vertex or edge?

All six of these problems are parameterized by a class of graphs $\mathcal{G}$ — for the four marginals problems, this is the graph induced by the given pairwise marginals, while for the models problems, it is the graph of the given Markov network. All of our reductions are of the form "for every class $\mathcal{G}$, if there is a polynomial-time algorithm for solving inference problem $B$ for (model) graphs in $\mathcal{G}$, then there is a polynomial-time algorithm for marginals problem $A$ for (marginal) graphs in $\mathcal{G}$" — that is, $A$ reduces to $B$. Our main results, which are summarized in Figure 1, can be stated informally as follows:

- CONSISTENCY reduces to MAP INFERENCE.
- CLOSEST CONSISTENCY reduces to MAP INFERENCE.
- SMALL SUPPORT reduces to MAP INFERENCE.
- MAX ENTROPY reduces to GENERALIZED PARTITION.[3]

While connections between some of these problems are known for *specific classes* of graphs — most notably in trees, where all of these problems are tractable and rely on common underlying algorithmic approaches such as dynamic programming — the novelty of our results is their generality, showing that the above reductions hold for *every* class of graphs.

All of our reductions share a common and powerful technique: the use of the ellipsoid method for Linear Programming (LP), with the key step being the articulation of an appropriate separation oracle. The first three problems we consider have a straightforward LP formulation which will typically have a number of variables that is equal to the number of joint settings, and therefore exponential in the number of variables; for the MAX ENTROPY problem, there is an analogous convex program formulation. Since our goal is to run in time polynomial in the input length (the number and size of given marginals), the straightforward LP formulation will not suffice. However, by passing to the dual LP, we instead obtain an LP with only a polynomial number of variables, but an exponential number of constraints that can be represented implicitly. For each of the reductions above, we show that the required separation oracle for these implicit constraints is provided exactly by the corresponding inference problem (MAP INFERENCE or GENERALIZED PARTITION). We believe this technique may be of independent interest and have other applications in probabilistic inference.

It is perhaps surprising that in the study of problems strictly addressing properties of given marginals (which have received relatively little attention in the graphical models literature historically), problems of inference in full joint models (which have received great attention) should arise so naturally and generally. For the marginal problems, our reductions (via the ellipsoid method) effectively create a series of "fictitious" Markov networks such that the solutions to corresponding inference problems (MAP INFERENCE and GENERALIZED PARTITION) indirectly lead to a solution to the original marginal problems.

***Related Work:*** The literature on graphical models and probabilistic inference is rife with connections between some of the problems we study here for specific classes of graphical models (such as trees or otherwise sparse structures), and under specific algorithmic approaches (such as dynamic programming or message-passing algorithms more generally, and various forms of variational inference); see [2, 3, 4] for good overviews. In contrast, here we develop general and efficient reductions between marginal and inference problems that hold regardless of the graph structure or algorithmic approach; we are not aware of prior efforts in this vein. Some of the problems we consider are also either new or have been studied very little, such as CLOSEST CONSISTENCY and SMALL SUPPORT.

The CONSISTENCY problem has been studied before as the membership problem for the marginal polytope. In particular, [8] shows that finding the MAP assignment for Markov random fields with pairwise potentials can be cast as an integer linear program over the marginal polytope — that is, algorithms for the CONSISTENCY problem are useful subroutines for inference. Our work is the

first to show a converse, that inference algorithms are useful subroutines for decision and optimization problems for the marginal polytope. Furthermore, previous polynomial-time solutions to the CONSISTENCY problem generally give a compact (polynomial-size) description of the marginal polytope. Our approach dodges this ambitious requirement, in that it only needs a polynomial-time separation oracle (which, for this problem, turns out to be MAP inference). As there are many combinatorial optimization problems with no compact LP formulation that admit polynomial-time ellipsoid-based algorithms — like non-bipartite matching, with its exponentially many odd cycle inequalities — our approach provides a new way of identifying computationally tractable special cases of problems concerning marginals.

The previous work that is perhaps most closely related in spirit to our interests are [5] and [6, 7]. These works provide reductions of some form, but not ones that are both general (independent of graph structure) and polynomial time. However, they do suggest both the possibility and interest in such stronger reductions. The paper [5] discusses and provides heuristic reductions between MAP INFERENCE and GENERALIZED PARTITION.

The work in [6, 7] makes the point that maximizing entropy subject to an (approximate) consistency condition yields a distribution that can be represented as a Markov network over the graph induced by the original data or marginals. As far as we are aware, however, there has been essentially no formal complexity analysis (i.e., worst-case polynomial-time guarantees) for algorithms that compute max-entropy distributions.[4]

## 2 Preliminaries

### 2.1 Problem Definitions

For clarity of exposition, we focus on the pairwise case in which every marginal involves at most two variables.[5] Denote the underlying random variables by $X_1, \ldots, X_n$, which we assume have range $[k] = \{0, 1, 2, \ldots, k\}$. The input is at most one real-valued *single marginal value* $\mu_{is}$ for every variable $i \in [n]$ and value $s \in [k]$, and at most one real-valued *pairwise marginal value* $\mu_{ijst}$ for every ordered variable pair $i, j \in [n] \times [n]$ with $i < j$ and every pair $s, t \in [k]$. Note that we allow a marginal to be only partially specified. The *data graph* induced by a set of marginals has one vertex per random variable $X_i$, and an undirected edge $(i, j)$ if and only if at least one of the given pairwise marginal values involves the variables $X_i$ and $X_j$. Let $M_1$ and $M_2$ denote the sets of indices $(i, s)$ and $(i, j, s, t)$ of the given single and pairwise marginal values, and $m = |M_1| + |M_2|$ the total number of marginal values. Let $A = [k]^n$ denote the space of all possible variable assignments. We say that the given marginals $\mu$ are *consistent* if there exists a (joint) probability distribution consistent with all of them (i.e., that induces the marginals $\mu$).

With these basic definitions, we can now give formal definitions for the marginals problems we consider. Let $\mathcal{G}$ denote an arbitrary class of undirected graphs.

- CONSISTENCY ($\mathcal{G}$): Given marginals $\mu$ such that the induced data graph falls in $\mathcal{G}$, are they consistent?

- CLOSEST CONSISTENCY ($\mathcal{G}$): Given (possibly inconsistent) marginals $\mu$ such that the induced data graph falls in $\mathcal{G}$, compute the consistent marginals $\nu$ minimizing $||\nu - \mu||_1$.

- SMALL SUPPORT ($\mathcal{G}$): Given (consistent or inconsistent) marginals $\mu$ such that the induced data graph falls in $\mathcal{G}$, compute a distribution that has a polynomial-size support and marginals $\nu$ that minimize $||\nu - \mu||_1$.

- MAX ENTROPY ($\mathcal{G}$): Given (consistent or inconsistent) marginals $\mu$ such that the induced data graph falls in $\mathcal{G}$, compute the maximum entropy distribution that has marginals $\nu$ that minimize $||\nu - \mu||_1$.

It is important to emphasize that all of the problems above are "model-free", in that we do not assume that the marginals are consistent with, or generated by, any particular model (such as a Markov network). They are simply given marginals, or "data".

For each of these problems, our interest is in algorithms whose running time is polynomial in the size of the input $\mu$. The prospects for this depend strongly on the class $\mathcal{G}$, with tractability generally following for "nice" classes such as tree or tree-like graphs, and intractability for the most general cases. Our contribution is in showing a strong connection between tractability for these marginals problems and the following inference problems for *any* class $G$.

- MAP INFERENCE ($\mathcal{G}$): Given a Markov network whose graph falls in $\mathcal{G}$, find the maximum a posteriori (MAP) or most probable joint assignment.[6]
- GENERALIZED PARTITION: Given a Markov network whose graph falls in $\mathcal{G}$, compute the partition function or normalization constant for the full joint distribution, possibly after conditioning on the value of a single vertex or edge.[7]

## 2.2 The Ellipsoid Method for Linear Programming

Our algorithms for the CONSISTENCY, CLOSEST CONSISTENCY, and SMALL SUPPORT problems use linear programming. There are a number of algorithms that solve explictly described linear programs in time polynomial in the description size. Our problems, however, pose an additional challenge: the obvious linear programming formulation has size exponential in the parameters of interest. To address this challenge, we turn to the ellipsoid method [9], which can solve in polynomial time linear programs that have an exponential number of implicitly described constraints, provided there is a polynomial-time "separation oracle" for these constraints. The ellipsoid method is discussed exhaustively in [10, 11]; we record in this section the facts necessary for our results.

**Definition 2.1 (Separation Oracle)** *Let* $\mathcal{P} = \{\mathbf{x} \in \mathbb{R}^n : \mathbf{a}_1^T \mathbf{x} \le b_1, \ldots, \mathbf{a}_m^T \mathbf{x} \le b_m\}$ *denote the feasible region of* $m$ *linear constraints in* $n$ *dimensions. A* separation oracle *for* $\mathcal{P}$ *is an algorithm that takes as input a vector* $\mathbf{x} \in \mathbb{R}^n$, *and either (i) verifies that* $\mathbf{x} \in \mathcal{P}$; *or (ii) returns a constraint* $i$ *such that* $\mathbf{a}_i^t \mathbf{x} > b_i$. *A* polynomial-time *separation oracle runs in time polynomial in* $n$, *the maximum description length of a single constraint, and the description length of the input* $\mathbf{x}$.

One obvious separation oracle is to simply check, given a candidate solution $\mathbf{x}$, each of the $m$ constraints in turn. More interesting and relevant are constraint sets that have size exponential in the dimension $n$ but admit a polynomial-time separation oracle.

**Theorem 2.2 (Convergence Guarantee of the Ellipsoid Method [9])** *Suppose the set* $\mathcal{P} = \{\mathbf{x} \in \mathbb{R}^n : \mathbf{a}_1^T \mathbf{x} \le b_1, \ldots, \mathbf{a}_m^T \mathbf{x} \le b_m\}$ *admits a polynomial-time separation oracle and* $\mathbf{c}^T \mathbf{x}$ *is a linear objective function. Then, the ellipsoid method solves the optimization problem* $\{\max \mathbf{c}^T \mathbf{x} : \mathbf{x} \in \mathcal{P}\}$ *in time polynomial in* $n$ *and the maximum description length of a single constraint or objective function. The method correctly detects if* $\mathcal{P} = \emptyset$. *Moreover, if* $\mathcal{P}$ *is non-empty and bounded, the ellipsoid method returns a vertex of* $\mathcal{P}$.[8]

Theorem 2.2 provides a general reduction from a problem to an intuitively easier one: *if* the problem of verifying membership in $\mathcal{P}$ can be solved in polynomial time, *then* the problem of optimizing an arbitrary linear function over $\mathcal{P}$ can also be solved in polynomial time. This reduction is "many-to-one," meaning that the ellipsoid method invokes the separation oracle for $\mathcal{P}$ a large (but polynomial) number of times, each with a different candidate point $\mathbf{x}$. See Appendix A.1 for a high-level description of the ellipsoid method and [10, 11] for a detailed treatment.

The ellipsoid method also applies to convex programming problems under some additional technical conditions. This is discussed in Appendix A.2 and applied to the MAX ENTROPY problem in Appendix A.3.

# 3 CONSISTENCY **Reduces to** MAP INFERENCE

The goal of this section is to reduce the CONSISTENCY problem for data graphs in the family $\mathcal{G}$ to the MAP INFERENCE problem for networks in $\mathcal{G}$.

**Theorem 3.1 (Main Result 1)** *Let $\mathcal{G}$ be a set of graphs. If the the* MAP INFERENCE $(\mathcal{G})$ *problem can be solved in polynomial time, then the* CONSISTENCY $(\mathcal{G})$ *problem can be solved in polynomial time.*

We begin with a straightforward linear programming formulation of the CONSISTENCY problem.

**Lemma 3.2 (Linear Programming Formulation)** *An instance of the* CONSISTENCY *problem admits a consistent distribution if and only if the following linear program (P) has a solution:*

$$(P) \quad \max_{\mathbf{p}} \qquad \qquad 0$$

*subject to:*

$$\sum_{a \in A : a_i = s} p_a = \mu_{is} \qquad \text{for all } (i, s) \in M_1$$
$$\sum_{a \in A : a_i = s, a_j = t} p_a = \mu_{ijst} \quad \text{for all } (i, j, s, t) \in M_2$$
$$\sum_{a \in A} p_a = 1$$
$$p_a \geq 0 \qquad \qquad \text{for all } a \in A.$$

Solving (P) using the ellipsoid method (Theorem 2.2), or any other linear programming method, requires time at least $|A| = (k+1)^n$, the number of decision variables. This is generally exponential in the size of the input, which is proportional to the number $m$ of given marginal values.

A ray of hope is provided by the fact that the number of *constraints* of the linear program in Lemma 3.2 is equal to the number of marginal values. With an eye toward applying the ellipsoid method (Theorem 2.2), we consider the dual linear program. We use the following notation. Given a vector $\mathbf{y}$ indexed by $M_1 \cup M_2$, we define

$$\mathbf{y}(a) = \sum_{(i,s) \in M_1 \,:\, a_i = s} y_{is} + \sum_{(i,j,s,t) \in M_2 \,:\, a_i = s, a_j = t} y_{ijst} \qquad (1)$$

for each assignment $a \in A$, and

$$\mu^T \mathbf{y} = \sum_{(i,s) \in M_1} \mu_{is} y_{is} + \sum_{(i,j,s,t) \in M_2} \mu_{ijst} y_{ijst}. \qquad (2)$$

Strong linear programming duality implies the following.

**Lemma 3.3 (Dual Linear Programming Formulation)** *An instance of the* CONSISTENCY *problem admits a consistent distribution if and only if the optimal value of the following linear program (D) is 0:*

$$(D) \quad \max_{\mathbf{y}, z} \qquad \mu^T \mathbf{y} + z$$

*subject to:*

$$\mathbf{y}(a) + z \leq 0 \qquad \text{for all } a \in A$$
$$\mathbf{y}, z \text{ unrestricted.}$$

The number of variables in (D) — one per constraint of the primal linear program — is polynomial in the size of the CONSISTENCY input.

What use is the MAP INFERENCE problem for solving the CONSISTENCY problem? The next lemma forges the connection.

**Lemma 3.4 (Map Inference as a Separation Oracle)** *Let $\mathcal{G}$ be a set of graphs and suppose that the* MAP INFERENCE $(\mathcal{G})$ *problem can be solved in polynomial time. Consider an instance of the* CONSISTENCY *problem with a data graph in $\mathcal{G}$, and a candidate solution $\mathbf{y}, z$ to the corresponding*

*dual linear program (D). Then, there is a polynomial-time algorithm that checks whether or not there is an assignment $a \in A$ that satisfies*

$$\sum_{(i,s) \in M_1 \,:\, a_i = s} y_{is} + \sum_{(i,j,s,t) \in M_2 \,:\, a_i = s, a_j = t} y_{ijst} > -z, \tag{3}$$

*and produces such an assignment if one exists.*

*Proof:* The key idea is to interpret $\mathbf{y}$ as the log-potentials of a Markov network. Precisely, construct a Markov network $N$ as follows. The vertex set $V$ and edge set $E$ correspond to the random variables and edge set of the data graph of the CONSISTENCY instance. The potential function at a vertex $i$ is defined as $\phi_i(s) = \exp\{y_{is}\}$ for each value $s \in [k]$. The potential function at an edge $(i,j)$ is defined as $\phi_{ij}(s,t) = \exp\{y_{ijst}\}$ for $(s,t) \in [k] \times [k]$. For a missing pair $(i,s) \notin M_1$ or 4-tuple $(i,j,s,t) \notin M_2$, we define the corresponding potential value $\phi_i(s)$ or $\phi_{ij}(st)$ to be 1. The underlying graph of $N$ is the same as the data graph of the given CONSISTENCY instance and hence is a member of $\mathcal{G}$.

In the distribution induced by $N$, the probability of an assignment $a \in [k]^n$ is, by definition, proportional to

$$\left( \prod_{i \in V \,:\, (i,a_i) \in M_1} \exp\{y_{ia_i}\} \right) \left( \prod_{(i,j) \in E \,:\, (i,j,a_i,a_j) \in M_2} \exp\{y_{ija_i a_j}\} \right) \;=\; \exp\{\mathbf{y}(a)\}.$$

That is, the MAP assignment for the Markov network $N$ is the assignment that maximizes the left-hand size of (3).

Checking if some assignment $a \in A$ satisfies (3) can thus be implemented as follows: compute the MAP assignment $a^*$ for $N$ — by assumption, and since the graph of $N$ lies in $\mathcal{G}$, this can be done in polynomial time; return $a^*$ if it satisfies (3), and otherwise conclude that no assignment $a \in A$ satisfies (3). ∎

All of the ingredients for the proof of Theorem 3.1 are now in place.

*Proof of Theorem 3.1:* Assume that there is a polynomial-time algorithm for the MAP INFERENCE $(\mathcal{G})$ problem with the family $\mathcal{G}$ of graphs, and consider an instance of the CONSISTENCY problem with data graph $G \in \mathcal{G}$. Deciding whether or not this instance has a consistent distribution is equivalent to solving the program (D) in Lemma 3.3. By Theorem 2.2, the ellipsoid method can be used to solve (D) in polynomial time, provided the constraint set admits a polynomial-time separation oracle. Lemma 3.4 shows that the relevant separation oracle is equivalent to computing the MAP assignment of a Markov network with graph $G \in \mathcal{G}$. By assumption, the latter problem can be solved in polynomial time. ∎

We defined the CONSISTENCY problem as a decision problem, where the answer is "yes" or no." For instances that admit a consistent distribution, we can also ask for a succinct representation of a distribution that witnesses the marginals' consistency. We next strengthen Theorem 3.1 by showing that for consistent instances, under the same hypothesis, we can compute a small-support consistent distribution in polynomial time. See Figure 2 for the high-level description of the algorithm.

**Theorem 3.5 (Small-Support Witnesses)** *Let $\mathcal{G}$ be a set of graphs. If the MAP INFERENCE $(\mathcal{G})$ problem can be solved in polynomial time, then for every consistent instance of the CONSISTENCY $(\mathcal{G})$ problem with $m = |M_1| + |M_2|$ marginal values, a consistent distribution with support size at most $m + 1$ can be computed in polynomial time.*

*Proof:* Consider a consistent instance of CONSISTENCY with data graph $G \in \mathcal{G}$. The algorithm of Theorem 3.1 concludes by solving the dual linear program of Lemma 3.3 using the ellipsoid method. This method runs for a polynomial number $K$ of iterations, and each iteration generates one new inequality. At termination, the algorithm has identified a "reduced dual linear program", in which a set of only $K$ out of the original $(k+1)^n$ constraints is sufficient to prove the optimality of its solution. By strong duality, the corresponding "reduced primal linear program," obtained from the linear program in Lemma 3.2 by retaining only the decision variables corresponding to the $K$

1. Solve the dual linear program (D) (Lemma 3.3) using the ellipsoid method (Theorem 2.2), using the given polynomial-time algorithm for MAP INFERENCE ($\mathcal{G}$) to implement the ellipsoid separation oracle (see Lemma 3.4).

2. If the dual (D) has a nonzero (and hence, unbounded) optimal objective function value, then report "no consistent distributions" and halt.

3. Explicitly form the reduced primal linear program (P-red), obtained from (P) by retaining only the variables that correspond to the dual inequalities generated by the separation oracle in Step 1.

4. Solve (P-red) using a polynomial-time linear programming algorithm that returns a vertex solution, and return the result.

Figure 2: High-level description of the polynomial-time reduction from CONSISTENCY ($\mathcal{G}$) to MAP INFERENCE ($\mathcal{G}$) (Steps 1 and 2) and postprocessing to extract a small-support distribution that witnesses consistent marginals (Steps 3 and 4).

reduced dual constraints, has optimal objective function value 0. In particular, this reduced primal linear program is feasible.

The reduced primal linear program has a polynomial number of variables and constraints, so it can be solved by the ellipsoid method (or any other polynomial-time method) to obtain a feasible point $\mathbf{p}$. The point $\mathbf{p}$ is an explicit description of a consistent distribution with support size at most $K$. To improve the support size upper bound from $K$ to $m + 1$, recall from Theorem 2.2 that $\mathbf{p}$ is a vertex of the feasible region, meaning it satisfies $K$ linearly independent constraints of the reduced primal linear program with equality. This linear program has at most one constraint for each of the $m$ given marginal values, at most one normalization constraint $\sum_{a \in A} p_a = 1$, and non-negativity constraints. Thus, at least $K - m - 1$ of the constraints that $\mathbf{p}$ satisfies with equality are non-negativity constraints. Equivalently, it has at most $m + 1$ strictly positive entries. ∎

## 4 CLOSEST CONSISTENCY, SMALL SUPPORT **Reduce to** MAP INFERENCE

This section considers the CLOSEST CONSISTENCY and SMALL SUPPORT problems. The input to these problems is the same as in the CONSISTENCY problem — single marginal values $\mu_{is}$ for $(i, s) \in M_1$ and pairwise marginal values $\mu_{ijst}$ for $(i, j, s, t) \in M_2$. The goal is to compute sets of marginals $\{\nu_{is}\}_{M_1}$ and $\{\nu_{ijst}\}_{M_2}$ that are consistent and, subject to this constraint, minimize the $\ell_1$ norm $||\mu - \nu||_1$ with respect to the given marginals. An algorithm for the CLOSEST CONSISTENCY problem solves the CONSISTENCY problem as a special case, since a given set of marginals is consistent if and only if the corresponding CLOSEST CONSISTENCY problem has optimal objective function value 0. Despite this greater generality, the CLOSEST CONSISTENCY problem also reduces in polynomial time to the MAP INFERENCE problem, as does the still more general SMALL SUPPORT problem.

**Theorem 4.1 (Main Result 2)** *Let $\mathcal{G}$ be a set of graphs. If the* MAP INFERENCE *($\mathcal{G}$) problem can be solved in polynomial time, then the* CLOSEST CONSISTENCY *($\mathcal{G}$) problem can be solved in polynomial time. Moreover, a distribution consistent with the optimal marginals with support size at most $3m + 1$ can be computed in polynomial time, where $m = |M_1| + |M_2|$ denotes the number of marginal values.*

The formulation of the CLOSEST CONSISTENCY ($\mathcal{G}$) problem has linear constraints — the same as those in Lemma 3.2, except with the given marginals $\mu$ replaced by the computed consistent marginals $\nu$ — but a nonlinear objective function $||\mu - \nu||_1$. We can simulate the absolute value functions in the objective by adding a small number of variables and constraints. We provide details and the proof of Theorem 4.1 in Appendix A.4.

## Footnotes

[1] For a simple example, consider three random variables for which each pairwise marginal specifies that the settings (0,1) and (1,0) each occurs with probability $1/2$. The corresponding "data graph" is a triangle. This requires that each variable always disagrees with the other two, which is impossible.

[2] For example, consider random variables $X, Y, Z$. Suppose the pairwise marginals for $X$ and $Y$ and for $Y$ and $Z$ specify that all four binary settings are equally likely. No pairwise marginals for $X$ and $Z$ are given, so the data graph is a two-hop path. One consistent distribution flips a fair coin independently for each variable; but another flips one coin for $X$, a second for $Y$, and sets $Z = X$. The former maximizes entropy while the latter minimizes support size.

[3]The conceptual ideas in this reduction are well known. We include a formal treatment in the Appendix for completeness and to provide an analogy with our other reductions, which are our more novel contributions.

[4]There are two challenges to doing this. The first, which has been addressed in previous work, is to circumvent the exponential number of decision variables via a separation oracle. The second, which does not seem to have been previously addressed, is to bound the diameter of the search space (i.e., the magnitude of the optimal Lagrange variables). Proving this requires using special properties of the MAX ENTROPY problem, beyond mere convexity. We adapt recent techniques of [13] to provide the necessary argument.

[5]All of our results generalize to the case of higher-order marginals in a straightforward manner.

[6]Formally, the input is a graph $G = (V, E)$ with a log-potential $\log \phi_i(s)$ and $\log \phi_{ij}(s, t)$ for each vertex $i \in V$ and edge $(i, j) \in E$, and each value $s \in [k] = \{0, 1, 2 \ldots, k\}$ and pair $s, t \in [k] \times [k]$ of values. The MAP assignment maximizes $P(a) := \prod_{i \in V} \phi_i(a_i) \prod_{(i,j) \in E} \phi_{ij}(a_i, a_j)$ over all assignments $a \in [k]^V$.

[7]Formally, given the log-potentials of a Markov network, compute $\sum_{a \in [k]^n} P(a)$; $\sum_{a : a_i = s} P(a)$ for a given $i, s$; or $\sum_{a : a_i = s, a_j = t} P(a)$ for a given $i, j, s, t$.

[8]A *vertex* is a point of $\mathcal{P}$ that satisfies with equality $n$ linearly independent constraints.

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
