[Supplementary Material · RoughgardenKearnsSupplementary.pdf]

# A Appendix: Supplementary Material

## A.1 High-Level Description of the Ellipsoid Method

Very roughly, the method works as follows. First, the optimization problem is reduced to a feasibility problem by adding the constraint $\mathbf{c}^T\mathbf{x} \geq Z$ to $\mathcal{P}$ and binary searching on $Z$ in an outer loop. For the feasibility problem, the method maintains an ellipsoid $E$ that is an outer bound (i.e., superset of) $\mathcal{P}$. At every iteration $i$, the method uses the separation oracle to check if the centroid of the current ellipsoid $E_i$ is feasible. If so, the method stops. If not, the separation oracle furnishes a violated constraint; intersecting this with $E_i$ yields a " partial ellipsoid" $E_i'$ that excludes the centroid of $E_i$. The next ellipsoid $E_{i+1}$ is taken to be the minimum-volume one that encloses $E_i'$. A technical argument shows that the volume of $E_{i+1}$ is significantly less than that of $E_i$, and this leads to the convergence bound. See [10, 11] for details.

## A.2 The Ellipsoid Method for Convex Programming

As is well known, the MAX ENTROPY problem is a convex optimization problem. The ellipsoid method can be adapted to such problems under mild technical conditions. The following guarantee, which can be derived from more general results (e.g. [10, 11]), is sufficient for our purposes. It states that, provided the relevant magnitudes of feasible solutions and objective function values are at most exponential, then given polynomial-time oracles for evaluating the objective function of a convex program and its gradient, the ellipsoid method can solve the program in time polynomial in the input size and the desired precision.

**Theorem A.1 (The Ellipsoid Method for Convex Programs)** *Consider an unconstrained minimization problem of the form $\{\inf_{\mathbf{y}} f(\mathbf{y}) : \mathbf{y} \in \mathbb{R}^n\}$ and suppose that:*

1. *For an a priori known bound $R$, there is an optimal solution $\mathbf{y}^*$ with $||\mathbf{y}||_\infty \leq R$.*

2. *For every pair $\mathbf{y}^1, \mathbf{y}^2$ with $||\mathbf{y}^1||_\infty, ||\mathbf{y}^2||_\infty \leq R$, $|f(\mathbf{y}^1) - f(\mathbf{y}^2)| \leq K$.*

3. *For every point $\mathbf{y}$, the objective function value $f(\mathbf{y})$ and its gradient $\nabla f(\mathbf{y})$ can be evaluated in time polynomial in $n$ and description length of $\mathbf{y}$.*

*Then, given $\epsilon > 0$, the ellipsoid method computes a point $\tilde{\mathbf{y}}$ such that $f(\tilde{\mathbf{y}}) < f(\mathbf{y}^*) + \epsilon$ in time polynomial in $n$, $\log R$, $\log K$, and $\log \frac{1}{\epsilon}$.*

## A.3 MAX ENTROPY **Reduces to** GENERALIZED PARTITION

This section describes a reduction from the MAX ENTROPY $(\mathcal{G})$ problem to the GENERALIZED PARTITION $(\mathcal{G})$ problem that is based on the ellipsoid method for convex programming (Theorem A.1). Before explaining the precise technical conditions that enable this reduction, we first ignore computational complexity issues and review some well-known theory about the MAX ENTROPY $(\mathcal{G})$ problem (see e.g. [12, 3] for more details).

The standard convex programming formulation of the MAX ENTROPY problem, subject to marginals $\{\mu\}$, is simply the linear program (P) of Lemma 3.2, augmented with the entropy objective function:

$$(P - ME) \quad \sup_{\mathbf{p}} \quad \sum_{a \in A} p_a \ln \frac{1}{p_a}$$

$$\text{subject to:}$$

$$\sum_{a \in A: a_i = s} p_a = \mu_{is} \quad \text{for all } (i, s) \in M_1$$

$$\sum_{a \in A: a_i = s, a_j = t} p_a = \mu_{ijst} \quad \text{for all } (i, j, s, t) \in M_2$$

$$\sum_{a \in A} p_a = 1$$

$$p_a \geq 0 \quad \text{for all } a \in A.$$

Using the notation in (1) and (2), we can write the dual program to (P-ME) as:

$$(D-ME) \quad \inf_{\mathbf{y}} \quad \mu^T \mathbf{y} + \ln \sum_{a \in A} \exp\{-\mathbf{y}(a)\}$$

subject to:

$$\mathbf{y} \text{ unrestricted.}$$

Assuming there is a feasible solution to (P-ME) with full support (i.e., $p_a > 0$ for all $a \in A$, a form of the Slater condition), strong duality holds and both convex programs have identical optimal objective function values. Moreover, in this case the maximum entropy distribution can be exactly represented as a Markov network $N$ with underlying graph equal to the data graph $G$ of the marginals $\mu$, with the negative of the log-potentials of $N$ corresponding to the optimal dual solution, similar to the mapping in the proof of Lemma 3.4. (Missing marginals from $\mu$ are defined to have zero log-potential.)

Under what conditions can we implement this approach with an algorithm with running time polynomial in the size of the MAX ENTROPY input? To see why convexity is not obviously enough, observe that the number of decision variables in the programs (P-ME) and (D-ME) is proportional to the number $(k+1)^n$ of variable assignments, with is typically exponential in the input size. We can, however, apply the ellipsoid method (Theorem A.1) to the dual program (D-ME) provided conditions 1.-3. are met by the problem. The next lemma connects the third condition in Theorem A.1 to the GENERALIZED PARTITION problem.

**Lemma A.2 (Generalized Partition as a Gradient Oracle)** *Let $\mathcal{G}$ be a set of graphs and suppose that the GENERALIZED PARTITION ($\mathcal{G}$) problem can be solved in polynomial time. Consider an instance of the MAX ENTROPY problem with a data graph $G \in \mathcal{G}$, with corresponding dual convex program (D-ME). Then, there are algorithms for evaluating the objective function $f$ of (D-ME) and its gradient $\nabla f$ that run in time polynomial in the size of the MAX ENTROPY instance and the magnitude of the evaluation point $\mathbf{y}$.*

*Proof (sketch):* Consider an evaluation point $\mathbf{y}$. Let $N$ denote the Markov network for which $\mathbf{y}$ is the negative of the log-potentials, as above. The graph of $N$ is identical to the data graph $G$ of the given MAX ENTROPY instance. The term $\sum_a \exp\{-\mathbf{y}(a)\}$ in the objective function of (D-ME) is precisely the Partition function of $N$. Given this quantity, which by assumption can be computed in polynomial time, the rest of the objective function is straightforward to compute.

Second, a simple computation shows that the gradient component $\nabla f_{is}(\mathbf{y})$ at $\mathbf{y}$ corresponding to $(i, s) \in M_1$ is

$$\mu_{is} - \frac{\sum_{a \,:\, a_i = s} \mathbf{y}(a)}{\sum_{a \in A} \mathbf{y}(a)},$$

and similarly for components of the form $\mu_{ijst}$. There are a polyomial number of components, and each is straightforward to compute given the assumed polynomial-time algorithm for the GENERALIZED PARTITION problem. ∎

Singh and Vishnoi [13] studied entropy maximization over combinatorial structures subject to single marginals, and in their context identified an additional condition, essentially a quantitative strengthening of the Slater condition, that implies the first two conditions of Theorem A.1 for the dual program (D-ME). To adapt it to our settings, consider a set $\mu$ of marginal values defined on $M_1 \cup M_2$. Let $\mathcal{P}(M_1, M_2)$ denote the set of all marginal vectors $\nu$ induced by probability distributions over $A$. For example, a set of marginals $\mu$ is consistent if and only if $\mu \in \mathcal{P}(M_1, M_2)$. More strongly, we say that $\mu$ is $\eta$-*strictly feasible* if the intersection of the ball with center $\mu$ and radius $\eta$ with the set $\mathcal{P}(M_1, M_2)$ is contained in the relative interior of $\mathcal{P}(M_1, M_2)$. Our results below are interesting when $\eta$ is at least some inverse polynomial function of the input size.

Following the proof in [13, Theorem 2.7] shows that, in every $\eta$-strictly feasible instance of MAX ENTROPY, there is an optimal dual solution $\mathbf{y}^*$ to (D-ME) with $||\mathbf{y}||_\infty \leq \frac{m}{\eta}$, where $m = |M_1| + |M_2|$ is the number of marginal values. We can therefore take the constant $R$ in Theorem A.1 to be $\frac{m}{\eta}$. Plugging this bound into the objective function of (D-ME) shows that we can take the constant $K$ in Theorem A.1 to be exponential in $\frac{m}{\eta}$. Applying Theorem A.1 then gives the following reduction from the MAX ENTROPY problem to the GENERALIZED PARTITION problem.

**Theorem A.3 (Main Result 3)** *Let $\mathcal{G}$ be a set of graphs. If the* GENERALIZED PARTITION $(\mathcal{G})$ *problem can be solved in polynomial time, then every $\eta$-strictly feasible instance of the* MAX ENTROPY $(\mathcal{G})$ *problem can be solved up to error $\epsilon$ in time polynomial in the input size, $\frac{1}{\eta}$, and $\log \frac{1}{\epsilon}$.*

## A.4 Proof of Theorem 4.1:
CLOSEST CONSISTENCY **and** SMALL SUPPORT **Reduce to** MAP INFERENCE

**Lemma A.4 (LP Formulation for** CLOSEST CONSISTENCY**)** *The consistent marginals $\{\nu\}$ that minimize the $\ell_1$ distance $||\nu - \mu||_1$ to the given marginals $\{\mu\}$ correspond to optimal solutions to the following linear program:*

$$(P - close) \quad \min_{\mathbf{p}, \nu, \sigma} \quad \sum_{(i,s) \in M_1} \sigma_{is} + \sum_{(i,j,s,t) \in M_2} \sigma_{ijst}$$

subject to:

$$
\begin{aligned}
\sigma_{i,s} &\geq \nu_{is} - \mu_{is} && \text{for all } (i,s) \in M_1 \\
\sigma_{i,s} &\geq \mu_{is} - \nu_{is} && \text{for all } (i,s) \in M_1 \\
\sigma_{i,j,s,t} &\geq \nu_{ijst} - \mu_{ijst} && \text{for all } (i,j,s,t) \in M_2 \\
\sigma_{i,j,s,t} &\geq \mu_{ijst} - \nu_{ijst} && \text{for all } (i,j,s,t) \in M_2 \\
\sum_{a \in A: a_i = s} p_a &= \nu_{is} && \text{for all } (i,s) \in M_1 \\
\sum_{a \in A: a_i = s, a_j = t} p_a &= \nu_{ijst} && \text{for all } (i,j,s,t) \in M_2 \\
\sum_{a \in A} p_a &= 1 \\
p_a &\geq 0 && \text{for all } a \in A.
\end{aligned}
$$

*Proof:* The constraints enforce the inequality $\sigma_{i,s} \geq |\nu_{i,s} - \mu_{i,s}|$ for all $(i,s) \in M_1$, and similarly for the pariwise constraints and $M_2$, at every feasible solution. The minimization objective ensures that equality holds for every such constraint at every optimal solution. Thus, optimal solutions to this linear program are in correpondence with those of the more straightforward nonlinear formulation. ∎

We next need to pass to the linear programming dual to (P-close) to enable application of the ellipsoid method. The negative of this dual is, after some simplifications, as follows.

$$(D - close) \quad \max_{\mathbf{y}, z} \quad \mu^T \mathbf{y} + z$$

subject to:

$$
\begin{aligned}
\mathbf{y}(a) + z &\leq 0 && \text{for all } a \in A \\
-1 \leq y_{is} &\leq 1 && \text{for all } (i,s) \in M_1 \\
-1 \leq y_{ijst} &\leq 1 && \text{for all } (i,j,s,t) \in M_2 \\
z & \text{ unrestricted.}
\end{aligned}
$$

Our proof of Theorem 4.1 now follows the same outline as that of Theorem 3.1.

*Proof of Theorem 4.1:* Assume that there is a polynomial-time algorithm for the MAP INFERENCE $(\mathcal{G})$ problem with the family $\mathcal{G}$ of graphs, and consider an instance of the CLOSEST CONSISTENCY problem with data graph in $G \in \mathcal{G}$. The ellipsoid method can be used to solve the dual linear program (D-close) in polynomial time, provided the constraint set admits a polynomial-time separation oracle. Given a candidate dual solution $\mathbf{y}, z$, the polynomially many constraints that enforce $|y| \leq 1$ can be checked explicitly, and the rest can be checked by computing the MAP assignment of a Markov network with graph $G$, as in Lemma 3.4. By assumption, this MAP inference problem can be solved in polynomial time.

To recover an optimal solution for the CLOSEST CONSISTENCY instance, and to solve the SMALL SUPPORT problem, we proceed as in the proof of Theorem 3.5. We form a reduced version of (P-close), with variables corresponding to the (polynomially many) inequalities of (D-close) that were generated by the ellipsoid method. This reduced linear program has the same optimal objective function value as (P-close) and has polynomial size. The algorithm concludes by returning an optimal solution of this reduced linear program. Since this linear program has at most $3m + 1$ constraints other than the non-negativity constraints, every optimal vertex solution has support size at most $3m + 1$. ∎