[Reviews · NeurIPS 2013]

Submitted by Assigned_Reviewer_4

This paper explores new problems in the theory of graphical models, and shows how to reduce these problems to MAP and marginal inference.

The basic problem the paper concerns itself with is consistency: given a set of marginals over some variables, is there a joint distribution which is consistent with these marginals? An example of a set of marginals which does not correspond to a valid joint distribution is any solution of an LP relaxation which is not tight. The paper shows how to pose the consistency problem as an LP with exponentially many variables (one per joint setting), and then how to dualize this LP and find a polynomial-time cutting planes algorithm where the oracle for finding the maximally violated constraint is equivalent to MAP inference in the graphical model induced by the marginals

This idea extends easily to successively more complex settings: it can be used to find the closest (in total variation distance) consistent marginal to a given marginal, or finding a joint distribution with small support which is closest to a given marginal.

The paper also claims that Max Entropy, finding the maximum-entropy joint distribution which is consistent to a given set of marginals is reducible to computing the partition function (i.e., doing marginal inference). However this is known, as the problem of finding the maximum entropy distribution given some constraints is equivalent to maximizing the likelihood of a dual formulation of the problem, which is roughly equivalent to the reduction presented in the appendix.

The paper appears to be correct: the proofs are simple and easy to follow, and so are the main arguments. However, no result in the paper is surprising, and it is not clear why these results are interesting, or how they can be useful in the future.

---

Post author response: the author response clarified substantially the motivation behind the paper, and made me lean more towards accepting it. I strongly recommend the authors to fold in their motivating paragraphs in the main text, as well as restructure it so it does not appear to be unfinished as it does now.
Summary: The paper explores some new problems in graphical models and shows how to reduce them to inference. However, the problems aren't that interesting, and not obviously useful.

Submitted by Assigned_Reviewer_5

*** UPDATE following author response ***

The reviewers feel the paper would be significantly strengthened by augmenting the paper with some of the additional motivation given in the author response.

The particular point that seems most promising to me is:

3. Our ellipsoid-based approach is novel and, we hope, will find further applications in the field. Previous polynomial-time solutions to CONSISTENCY generally give a compact (polynomial-size) description of the marginal polytope. Our approach dodges this ambitious requirement, in that it only needs a polynomial-time separation oracle (which, for this problem, turns out to be MAP inference). There are many combinatorial optimization problems with no compact LP formulation that admit polynomial-time ellipsoid-based algorithms (like nonbipartite matching, with its exponentially many odd cycle inequalities).

This makes it sound like there is a chance of finding new polynomial-time CONSISTENCY algorithms for a set of models where MAP is tractable but the marginal polytope does not have a compact representation. It would be great if the authors could spell this out.

---------------

The authors demonstrate polynomial-time reducibility from several
problems relating to consistency of pseudomarginals of a graphical
model to the problems of inference in the same graphical
model. Specifically, their main result is to show that testing whether
a given vector belongs to the marginal polytope of G is
polynomial-time reducible to the problem of MAP inference in a
graphical model over G. The reduction works by expressing the
consistency problem as an LP with exponentially many variables. The
dual LP, which has an exponential number of constraints, can be solved
in polynomial time by the ellipsoid method if the separation problem
can be solved in polynomial time. The required separation oracle is a
MAP inference problem over G, which shows the desired reduction. They
present related results that follow similar reasoning, e.g., for the
problem of finding the closest point in the marginal polytope to a
provided vector of pseudomarginals.

The paper is technically sound, but it seems incomplete---it ends
abruptly without actually presenting several of the results claimed in
the introduction. I get the sense the authors ran out of space, time,
or both. I took a look at the result in the appendix about MAX-ENTROPY
reducing to GENERALIZED-PARTITION. This looks completely standard to
me: (1) the dual of the maximum entropy problem (problem D-ME in the
paper) is that of maximizing likelihood in an exponential family, (2)
this is a convex problem, (3) it can be solved in polynomial time by
the ellipsoid algorithm if the log-likelihood function and its
gradient can be computed in polynomial time, (4) the log-likelihood
function is a linear function minus the log-partition function, so
computing it efficiently amounts to computing the log-partition
function efficiently. I don't see what the new contribution is.

Overall, the authors could do a much better job situating their work
in relation to previous work. My initial reaction was to guess that
these results would follow from existing results about the theory of
exponential family models (cf. Wainwright and Jordan [2]), which
already describes very strong connections between marginals and
models (e.g., a bijection between marginals and models in a minimal
exponential family). I no longer think the main result (Theorem 3.1)
follows from that literature, but it would help this paper
considerably if the authors could point out the connections and
differences. The related work does a very poor job of this
currently---it simply points to three standard books/monographs about
graphical models.

There are some specific connections to prior work that should be
mentioned. In particular, the CONSISTENCY problem is exactly that of
testing membership in the marginal polytope of G. The authors should
mention this explicitly, as much is known about the marginal polytope,
and the membership problem which they study has appeared before, and
is known to be NP-complete:

David Sontag and Tommi Jaakkola, On Iteratively Constraining the
Marginal Polytope for Approximate Inference and MAP, Tech. report,
2007

The paper is reasonably clear, but imprecise in a few places,
especially surrounding the problem definitions. In particular, the
definition of SMALL-SUPPORT is very informal and probably not correct
as stated. "Polynomial-size" can only be judged asymptotically; for
this case when you have a specific input instance in hand, you
probably need to require that the support size is bounded by a
specific function of the input size. Also, the MAX-ENTROPY problem as
stated seems to want to simulteneuously maximize entropy and minimize
the L1 distance from \mu to the marginal polytope. I doubt this is
possible. Perhaps you mean to maximize entropy subject to the
constraint of \nu being a minimizer of the L1 distance?

Regarding novelty and significance, there is really one primary
observation in this paper: that MAP inference is the dual separation
oracle for the LP formulation of the CONSISTENCY problem (Theorem
3.1). As far as I know, this is new. Several other results
(CLOSEST-CONSISTENCY and SMALL-SUPPORT) are slight variants that
follow from standard LP tricks. The main result is intriguing, but it
is hard to see its significance. The reductions don't point to new
algorithms or complexity results for problems that people care about
(i.e., CONSISTENCY does not arise in practice, and it does not imply
new hardness results for MAP inference). The paper would be
strengthened considerably if the authors could point to some
application of the results.

Minor comments
* The characterization of Markov networks and Bayesian networks on
line 31 seems like it is equating potentials and CPTs with "local
marginals". I don't agree with this characterization---potentials
are not marginals. The marginal inference problem is all about
converting from the potentials to marginals (e.g. [3]). I suggest
clarifying the wording here.
* Figure 1: the text is very hard to read.
* Line 121: the number of LP variables is *equal* to the number of
joint settings (which is exponential in the number of random
variables)
* Line 162: This wording/notation suggests that the marginal is the
scalar \mu_{is}. I suggest refining the notation (e.g. (\mu_{is}){s \in
[k]}) to be more precise, which will clarify that the marginal is
a vector. The same comment applies to \mu_{ijst}.
* Line 167: "Let M_1 and M_2 denote the indices (i,s) and
(i,j,s,t)". This is a bit vague. Can you clarify that M_1 and M_2
are *sets* of indices.
* Line 394: Do you mean CLOSEST-CONSISTENCY instead of CONSISTENCY
here?
* Line 401: mathematical programming --> mathematical program
Summary: The paper makes an interesting observation: MAP inference is the dual
separation oracle of the LP for testing membership in the marginal
polytope. However, the significance of this result is unclear, and the
paper is incomplete: it ends before presenting several of the main
results, and does not do a very good job relating to prior work.

Submitted by Assigned_Reviewer_6

This paper considers the well-known-to-be-difficult problem of determining if a given set of marginal distributions are "consistent" in the sense that there is some joint distribution that corresponds to them. (The local polytope and such enforce a set of necessary conditions only.) Conceptually, what is done is very simple and (to my eyes) clever.

The main idea is that one can set of a naive linear programming problem of optimizing a (exponentially large) joint distribution under a null objective under a set of (linear) marginalization constraints. This is totally impractical, but since the set of marginalization constraints is polynomial, if one takes the dual of this LP, one obtains a problem with a polynomial variable size, but exponentially many constraints. The crucial observation that drives the paper is that the problem of finding a violated constraint in this dual LP is equivalent to a MAP inference problem. Thus, one can solve consistency by starting with a relaxed dual LP, and repeatedly solving MAP problems to add constraints. (Of course, MAP in NP hard in the worst case, but there are many algorithms that work well in practice.) The paper further argues that this can be an instance of the Ellipsoid algorithm meaning that each generated constraint reduces the state space enough that only a polynomial number of constraints will be needed, assuming the problem is consistent. (I think I understand correctly that it could take exponential time to prove inconsistency-- the paper should be slightly more explicit on this point.)

Based on this observation, the Ellipsoid algorithm can solve consistency, and small modifications to the linear problem can be introduced (with a l1 / total variation objective) to find the *closest* set of consistent marginals.

One question I do wonder about is if the MAX-ENTROPY -> Partition theorem should really be considered part of the paper? I know it is in the additional material, but the main text doesn't even *state* the result.

small comment:
- missing a bar on line 396

Edit after author response:

I am in agreement with the other reviewers that the author response did a better job of spelling out the motivation than the paper itself. Thus, I'd encourage putting a similar discussion into the introduction of the paper, which may lead to the paper having an impact on a larger audience.
Summary: Interesting theoretical paper on an understudied problem, a clear accept.
Author Feedback

Author rebuttal: We thank all three reviewers for their helpful and fair remarks. We apologize for the length of the rebuttal:), but do believe it directly addresses the questions raised by the reviewers. The main concerns seem to center on the significance of our results and their relationship to what is already known. We agree that we should clarify these points, and in the event of acceptance we would do so in the specific ways outlined below.

Why the (CLOSEST) CONSISTENCY problem and our solution to it are interesting:

1. As one reviewer rightfully points out, the CONSISTENCY problem has already been well studied by the community, as the membership problem for the marginal polytope. For example, in the suggested Sontag-Jakkaola (2007) paper, the authors explain "Finding the MAP assignment for MRFs with pairwise potentials can be cast as an integer linear program over the marginal polytope." That is, algorithms for the CONSISTENCY problem are useful subroutines for inference.

2. As far as we know, our work is the first to show a converse, that inference algorithms are useful subroutines for decision and optimization problems for the marginal polytope.

3. Our ellipsoid-based approach is novel and, we hope, will find further applications in the field. Previous polynomial-time solutions to CONSISTENCY generally give a compact (polynomial-size) description of the marginal polytope. Our approach dodges this ambitious requirement, in that it only needs a polynomial-time separation oracle (which, for this problem, turns out to be MAP inference). There are many combinatorial optimization problems with no compact LP formulation that admit polynomial-time ellipsoid-based algorithms (like nonbipartite matching, with its exponentially many odd cycle inequalities).

4. The (CLOSEST) CONSISTENCY problem is also interesting in its own right. Is given (noisy) data consistent? If not, what is the minimal perturbation to the data necessary to recover coherence? If one or more marginals is missing, what range of values is consistent with the known marginals?

Why the SMALL SUPPORT problem is interesting:

1. When there are many consistent distributions, which one should be singled out? There are, of course, well-known reasons to prefer the maximum entropy distribution. But this is not the only approach. For example, consider the three features "voted Republican in 2012", "supports Obamacare", and "supports tougher gun control", and suppose the single marginals are .5, .5, .5. The max-entropy distribution is uniform over the 8 possibilities. We might expect reality to hew closer to a small support distribution, perhaps even 50/50 over the two vectors 100 and 011.

2. In general, it is interesting to compute a small set of "representative individuals" (100 and 011 in the example above) such that the data set is consistent with a suitable distribution over these individuals. This is precisely the SMALL SUPPORT problem.

3. The SMALL SUPPORT problem can be viewed as a tractable relaxation of the problems of computing the distribution that minimizes the entropy or support size (which, as concave minimization and l_0 minimization problems, we strongly suspect are NP-hard, even for simple graph structures).

4. An algorithm for SMALL SUPPORT can be used as a heuristic for data visualization in the following way. Suppose marginals \mu are consistent. By varying the objective function in Lemma 3.2 (i.e., the direction of optimization), multiple small support distributions consistent with the data can be generated. These can be regarded as a small number of "extreme cases", to be examined by a domain expert.

On the MAX ENTROPY problem:

1. We do not claim our solution to the MAX ENTROPY problem as a primary contribution of the paper. We agree that the main concepts applied in the solution are well known.

2. As far as we know, however, there has been essentially no formal complexity analysis (i.e., worst-case polynomial-time guarantees) for algorithms that compute max-entropy distributions. Such guarantees do NOT follow automatically from convexity, for two reasons. The first, which has been addressed in previous work by the community, is the exponential number of decision variables (circumvented via a suitable separation oracle). The second, which does not seem to have been previously addressed, is the need to bound the diameter of the search space (e.g., if the optimal Lagrange variables are too big, it will take too long to find them). Bounding the search space requires using special properties of the MAX ENTROPY problem, and the last two paragraphs of page 11 outline the necessary argument.

3. An additional motivation for including the MAX ENTROPY discussion in the paper is to provide an analogy to help interpret our main results, e.g. "marginal polytope membership reduces to (multiple invocations of) MAP inference, in a similar sense to how max-entropy computations reduce to generalized partition computations".